# Novel Antibacterial Approaches and Therapeutic Strategies

**DOI:** 10.3390/antibiotics14040404

**Published:** 2025-04-15

**Authors:** Gustavo A. Niño-Vega, Jorge A. Ortiz-Ramírez, Everardo López-Romero

**Affiliations:** Departamento de Biología, División de Ciencias Naturales y Exactas, Campus Guanajuato, Universidad de Guanajuato, Noria Alta s/n, col. Noria Alta, Guanajuato C.P. 36050, Mexico; jorge.armando.ortizrmz@gmail.com

**Keywords:** antibacterials, MDR, novel approaches

## Abstract

The increase in multidrug-resistant organisms worldwide is a major public health threat driven by antibiotic overuse, horizontal gene transfer (HGT), environmental drivers, and deficient infection control in hospitals. In this article, we discuss these factors and summarize the new drugs and treatment strategies suggested to combat the increasing challenges of multidrug-resistant (MDR) bacteria. New treatments recently developed involve targeting key processes involved in bacterial growth, such as riboswitches and proteolysis, and combination therapies to improve efficacy and minimize adverse effects. It also tackles the challenges of the Gram-negative bacterial outer membrane, stressing that novel strategies are needed to evade permeability barriers, efflux pumps, and resistance mechanisms. Other approaches, including phage therapy, AMPs, and AI in drug discovery, are also discussed as potential alternatives. Finally, this review points out the urgency for continued research and development (R&D), industry–academic partnerships, and financial engines to ensure that MDR microbes do not exceed the value of antibacterial therapies.

## 1. Introduction

The failure to effectively treat infectious diseases owing to an increasing trend in antimicrobial resistance (AMR) stands out as an emerging threat to public health, resulting in increased morbidity and mortality rates, extended hospitalization time, and greater financial burden. In a recent analysis, it was estimated that drug-resistant infections by 88 pathogen–drug combinations caused 4.95 million deaths worldwide, 1.27 million (25%) out of which were due to AMR [1] Particularly, the widespread use of antibiotics in human and veterinary medicine together with environmental pollution by residual antibacterial pharmaceuticals generate a serious concern in relation to the emergence and spread of multidrug-resistant (MDR) pathogens. Recently, the World Health Organization (WHO) has alerted the public of a “post-antibiotic era” where simple infections and minor injuries might once again become deadly threats [2] Moreover, antibiotics used to fight these infections can also have negative side effects [3], as the drugs used to eliminate the infectious agents also affect beneficial microbiota, leaving an urgent need to re-evaluate treatments in terms of their ability to specifically target pathogenic bacteria while conserving the necessary microbiota [3].

In hospitals, broad-spectrum antibacterials and invasive procedures contribute to the high prevalence of MDR microbes, particularly among immunocompromised patients, creating unique surveillance challenges. Medical devices like catheters and ventilators act as colonization sites for bacteria that can cause hospital-associated infections [4]. Moreover, the nature of resistant strains capable of forming biofilms should be considered, as they are hard to eliminate and lead to spread and repeated infections [5]. Horizontal gene transfer (HGT) adds even more complexity to this problem since it enables the sharing of resistant genes among bacterial species, thereby facilitating the spread of resistance in healthcare settings and ecosystems [6,7,8]. To overcome this, there is a need for novel therapeutic strategies with better long-term effects.

The exploration of new antibacterial agents is turning the attention toward previously unexplored bacterial mechanisms such as riboswitches and caseinolytic protease (ClpP) systems in the fight against bacterial resistance. Their goal is to inhibit those pathogen survival pathways at lower risk of developing resistance [9,10]. In addition, repurposing existing or investigational drugs approved for non-antibacterial use has been an effective strategy for discovering new therapies [11]. Combination therapies (pairing antibiotics with antivirulence agents—quorum-sensing inhibitors or efflux pump blockers) can also enhance the therapeutic effect [12,13]. This structural barrier has converted the development of effective drugs against pathogenic Gram-negative bacteria into a real challenge. Most antibiotics are limited by their ability to penetrate cells effectively because they face a protective outer membrane (OM) present in these bacteria. To overcome this challenge, novel approaches are under development to either alter and render the bacterial membrane more permeable or facilitate targeted drug uptake [14,15,16].

Advancements in genomic technologies have opened avenues for novel antibacterial drug discovery by improving the identification of targets within bacterial genomes. The rapid evolution of molecular biology and sequencing technologies, such as whole-genome sequencing (WGS) and metagenomics, have paved the way for identifying new drug targets, anticipating resistance patterns, and designing personalized therapies [17].

However, despite having made significant strides in antibacterial research, there has been a substantial decline in novel antibiotic development and a simultaneous rise in bacterial resistance over the past three decades [18]. The economic disincentives to the pharmaceutical industry have decreased its R&D investment in the field. Antibacterial treatments are for temporary use only, leading to reduced long-term profitability compared with medications for chronic diseases. Also, new government regulations and high clinical trial costs make it harder to enter the market. While initiatives, including public funding for early-stage research and market-entry rewards for antibacterial agents, have been suggested to incentivize development, they have not yet promoted sufficient investment in novel R&D within the industry [19,20].

Here, we review current antibacterial approaches, recent therapeutic developments, and some of the future strategies needed to fight the worldwide problem of AMR. Although each individual section presented here might serve as the subject of an in-depth review on its own, we consciously accepted the trade-off between breadth and depth in order to bridge these domains to offer readers a consolidated yet rigorous overview with an interdisciplinary perspective, inviting the broader scientific community to engage in these emerging solutions.

## 2. Factors Contributing to the Increase in Bacterial Infections

One of the major causes managed by human medicine is antibiotic overuse, which is widely documented as a driver of bacterial resistance to antibiotics. Overusing antibiotics for viral infections, the non-judicious use of broad-spectrum antibiotics, and patients not completing their antibiotic course all create selective pressure, giving resistant bacteria a survival advantage [6]. Even low doses of antibiotics can act as a selective pressure for adaptation in bacteria, raising the likelihood of mutations in resistance genes and expediting the emergence of MDR bacteria in healthcare settings [4]. Infections acquired in these environments are considered the third leading cause of human death after cancer and heart disease. Horizontal gene transfer (HGT) is the primary biological mechanism contributing to the rising resistance in bacteria and plays a major role in the spread of antibiotic resistance genes (ARGs). Mobile genetic elements serve as vehicles for HGT in both hospital and environmental settings, increasing the burden of resistance [5]. The most common way for HGT to disseminate ARGs in bacteria is conjugation. In the conjugation process, a donor bacterium can transfer plasmids containing its resistance genes directly into a recipient through contact between the two, especially in settings with high bacterial density, such as hospital or agricultural environments [6]. Conjugative plasmids transport several ARGs originating from MDR microbes [6]. Through transformation, bacteria can take up free DNA from their surroundings and acquire resistance genes. This frequently occurs in sites with a high abundance of dead bacterial cells, allowing ARGs to be taken up and incorporated into the genomes of living bacteria [7]. Another biological mechanism of HGT is transduction, where bacteriophages transport ARGs from one bacterium to another. Transduction is especially important in environments rich in bacteriophages, such as the human gut microbiome or wastewater treatment plants [8]. High levels of antibiotics favor the conjugation frequency and expression of plasmid-carried conjugation-related genes (as seen in wastewater or agricultural runoffs). Sub-minimal inhibitory concentration (MIC) levels of antimicrobials such as meropenem and ciprofloxacin have been shown to markedly facilitate the horizontal plasmid-mediated transfer of resistance from *Klebsiella pneumoniae* and *Escherichia coli* [21].

While we understand the significance of HGT on increased resistance, managing its effects is still challenging. Microbial communities are diverse in environments such as wastewater treatment plants, hospitals, and, most importantly, the human gut, making tracking and controlling ARG transfers out of complex microbiomes extremely difficult. In addition, biofilms on biotic and abiotic surfaces help resistant bacteria survive and spread due to the persistence of ARGs in biofilms.

The application of antibiotics to livestock generates pools of resistant bacteria that can transfer ARGs to human pathogens through HGT. Animal health products and antibiotics are widely used in livestock as growth promoters or to prevent infection, creating reservoirs of resistant bacteria. These resilient microorganisms can spread to humans through direct contact, food items, and environmental contamination. Antibiotic-resistant microbes can spread across water and soil through agricultural runoffs containing resistant bacteria and antibiotics [22]. Resistant bacteria reservoirs are made even more complex via migratory birds, soil, and air [23].

Nosocomial infections are considered the third leading cause of death, after cancer and cardiovascular disease. Catheters and ventilators are common sources of contamination from bacteria. Thus, bacteria-harboring resistance genes survive and disseminate rapidly to immunocompromised patients in such environments [4]. More recently, a study monitored the transfer of plasmids that harbored ARGs between bacterial species within a hospital setting [7]. This movement of plasmids between bacteria is a significant factor in the rapid spread of resistance in hospitals [7].

HGT is also facilitated to a significant extent by the presence of non-antibiotic compounds like heavy metals, food preservatives, and pharmaceuticals. Silver nanoparticles commonly used in personal care products release silver ions into the environmental system and induce oxidative stress response of bacteria, further stimulating the transfer of plasmid [24] and facilitating the conjugative transmission of ARGs. Likewise, well-known active pharmaceutical ingredients (APIs) such as the antidepressant fluoxetine and synthetic analgesics like acetaminophen promote ARG transfer with mechanisms involving increased permeability of the cell membrane and SOS response induction, activating DNA repair pathways that facilitate plasmid integration [21]. The increased membrane permeability, the stress responses, and an increased probability for intact HGT promote the dissemination of resistant microbes to environments beyond healthcare [25].

## 3. Search for Novel Targets and Therapies

Classical antibiotics act on essential bacterial processes, including cell wall synthesis (e.g., β-lactams), protein synthesis (e.g., aminoglycosides), and DNA replication (e.g., fluoroquinolones). Although effective, the rampant overprescription and abuse of these medications result in the emergence of resistant strains due to the selective pressure applied to the bacterial population owing to the transfer of the HGT elements [7]. Furthermore, such antibiotics commonly have toxicity and off-target effects when given in high doses or for a prolonged time, threatening their long-term effectiveness [26].

Instead, new antibiotics are based on the original bacterial processes that are important for survival and are less likely to contribute to the development of resistance. Riboswitches are non-coding RNA domains that detect specific target ligands and regulate gene expression in bacteria. A recent study targeting flavin mononucleotide (FMN), thiamine pyrophosphate (TPP), and glmS riboswitches with chimeric antisense oligonucleotides effectively resulted in the inhibition of bacterial growth but without exhibiting toxicity on human cell lines [9], representing a rational and selective action against bacteria such as *Staphylococcus aureus* and *Escherichia coli*, which are substantially resistant pathogens [10]. Antibiotics in development target the bacterial caseinolytic protease (ClpP), essential for maintaining proper levels of protein homeostasis within bacteria, promoting their death. This class of antibiotics, known as activators of self-compartmentalizing proteases (ACPs), has demonstrated activity against Gram-negative pathogens for which traditional antibiotics are ineffective, including the *Neisseria* species [10].

Combination therapies are one way to improve the effectiveness of both old and new antibiotics. The combined treatment of gallium (a siderophore quencher) and furanone C-30 (a quorum-sensing blocker) with the antibiotics colistin and ciprofloxacin has been demonstrated to restore sensitivity against *Pseudomonas aeruginosa* [12]. However, β-lactams are still ineffective in combatting resistant bacteria due to the production of β-lactamase by these microorganisms. The combination of meropenem–vaborbactam, consisting of a carbapenem and a β-lactamase inhibitor, can overcome this resistance mechanism and restore activity against carbapenem-resistant *Enterobacterales* [13].

Antibacterials that target bacterial membranes represent an alternative mode of action with the potential to avert resistance development. PQ401, a new class of neutral diarylurea compounds, displays broad-spectrum activity against antibiotic-resistant and antibiotic-tolerant *S. aureus*. PQ401 circumvents an antibiotic limitation where cross-resistance to membrane-active agents like polymyxins has been observed [27] by causing select disruption of bacterial membranes.

Novel antibiotics with less toxicity and improved safety profiles are being rolled out. The replacement of l-Val with d-Val significantly enhanced the stability and safety of engineered antimicrobial peptides (AMPs) by presenting potent activity against MDR bacteria in respiratory infections while exhibiting a 4-fold improvement in safety compared to earlier versions, showing less toxicity to human cells [28].

## 4. Challenges in Developing Antibiotics Targeting Gram-Negative Outer Membranes

### 4.1. Structural Differences Between the Cell Wall (CW) of Gram-Positive and Gram-Negative Bacteria

It is well documented that the cell wall (CW) of Gram-negative bacteria is much more complex than that of Gram-positive bacteria, which exhibits a bilayered envelope consisting of the cytoplasmic membrane and a thick peptidoglycan layer containing teichoic acids that interact with either the CW or the inner membrane. Gram-negative bacteria, on the other hand, are enveloped in a three-layered CW consisting of the inner membrane, a thin peptidoglycan layer, and an extra outer membrane (OM), a hydrophobic and impermeable layer that restricts drug permeability, thereby making Gram-negative bacteria more resistant to many antibiotics than Gram-positive bacteria [29].

### 4.2. Efflux Pumps and β-Lactamases

For an antibiotic to penetrate the CW of Gram-negative bacteria and reach an inhibitory concentration in the cytoplasm, it must cross the OM, which allows the passage of small hydrophilic but not large hydrophobic molecules. In addition to OM permeability restrictions, other mechanisms of resistance include efflux pumps (EPs) and β-lactamases, which evolved due to overuse and misuse of antibiotics as well as interbacterial gene transfer. These EPs are used by bacteria to expel diverse toxic compounds such as heavy metals, disinfectants, antiseptics, etc., and are currently considered a cause of bacterial pathogenesis, virulence, and biofilm formation. Furthermore, even when antibiotics penetrate the OM, Gram-negative bacteria can express these EPs, maintaining their intracellular concentration of antibiotics at a non-lethal level. Pumps in this group (e.g., the AcrAB-TolC system in *E. coli*) decrease the intracellular concentration of antibiotics, especially the lipophilic or bulkier ones, and contribute to multidrug resistance [15].

Typically, EPs are tripartite complexes spanning the inner membrane, the periplasmic space, and the OM [30]. Recently, EPs have been classified into six families according to their constituents and mechanisms of operation: SMR, MFS, ABC, PACE, MATE, and the more significant in primary resistance of MDR Gram-negative pathogens and challenging RND (resistance-nodulation cell division) superfamily [31]. Most compounds are optimized for Gram-positive pathogens and lack the required physicochemical properties for penetrating the OM of Gram-negative pathogens. New strategies attempt to broaden the chemical space by finding matrix molecules that can go in through the OM and not be extruded out by efflux pumps. Optimizing these parameters for higher permeability, coupled with the need to maintain optimal hydrophobicity and charge so the compound can pass through porins and enter into cells has been an avenue pursued in recent work [15]. Porins (e.g., *E. coli* OmpF) function as channels that allow for the passive diffusion of hydrophilic molecules, including some antibiotics, into the cell [32]. Nonetheless, porin mutations result in a decrease in expression and/or conformational alteration, limiting the entry of antibiotics. Furthermore, porins have dynamic open-closed states to control antibiotic permeability, and thus, drug molecules cannot reach target sites inside the cell stably [32].

β-Lactamases, on the other hand, break the β-lactam ring present in antibiotics such as penicillins, cephalosporins, monobactams, and carbapenems. The β-lactam ring interacts with penicillin-binding proteins (PBPs), inhibiting CW biogenesis and leading to bacterial death. These enzymes have been classified into groups A, B, and D, which comprise nucleophilic serine enzymes (SBLs), and E, which includes metallo β-lactamases (MBLs) [31,33]. These classes are differentially susceptible to several inhibitors. For instance, class A but not class B β-lactamases are sensitive to clavulanic acid and sulbactam, whereas A, C, and D groups are inhibited by avibactam [31]. Liu et al. [33] obtained a boronic acid-based compound, known as taniborbactam (20 VNRX-2133), active against SBLs and MBLs and carbapenem-resistant *P. aeruginosa* and Enterobacteriaceae. Meropenem–vaborbactam is a combination therapy of carbapenem meropenem with the β-lactamase inhibitor vaborbactam that inhibits serine carbapenemases and restores activity to meropenem for use against carbapenem-resistant *Enterobacterales*. It is indicated for complicated UTIs and is a last-resort therapy against carbapenem-resistant bacteria [34,35].

### 4.3. Strategies Based on Disrupting the OM

Despite being an efficient approach to overcoming resistance, antibiotics that act on essential players in the OM, for example, β-barrel assembly machinery (the BAM complex) and lipopolysaccharide(s) transport machinery (Lpt), are relatively new. The BAM complex inserts β-barrel proteins into the OM, whereas Lpt transports lipopolysaccharide (LPS) to the OM. These systems are also the target of inhibitors that permeabilize membranes and increase the susceptibility of bacteria to antibiotics [14]. A primary challenge in designing these inhibitors is that they must circumvent efflux pumps and reach high local concentrations at their sites of action. A range of compounds that repress OM assembly are non-soluble by efflux systems, necessitating combination procedures or adjustments to increase their intracellular retention [36].

A promising broad-spectrum antibacterial active against Gram-negative bacteria is darobactin A, which acts on the OM protein BamA, binding this target in the periplasm and impairing protein folding and integration. Thus, a major advantage of this antibacterial is to circumvent the need to pass over the permeability barrier [37]. It was recently shown that natural darobactin A, obtained from an entomopathogenic bacterium, can be processed to yield darobactins D (D22) and 69 (D69). D22 was active in vivo against murine infections of *Pseudomonas aeruginosa* and *E. coli* and rescued zebrafish embryos infected with *A. baumannii* [38].

OM disruptors block the synthesis of OM proteins or act by loosening the integrity of OM itself. In any case, they facilitate the entrance of antibiotics into the cell. Recently, a high number of compounds active against Gram-positive were analyzed for their activity against Gram-negative *E. coli* [31]. These authors highlighted two compounds that disrupted the OM of *E. coli* in synergy with linezolid and one that increased the potency of novobiocin [31]. Chemical perturbants such as MAC-0568743 disrupt the OM while selectively preserving the inner membrane, enabling antibiotics, such as vancomycin, to penetrate resistant Gram-negative bacteria [39]. These OM disruptors work in tandem with common antibiotics, improving their effectiveness against resistant strains. SPR741 is a novel polymyxin B analog that explicitly permeabilizes the outer membrane while leaving the inner membrane intact, enabling antibiotics to act on their intracellular targets more efficiently [40]. OM disruptors and Gram-positive active antibacterials have been reported to be effective against murine infections caused by *A. baumannii, Klebsiella pneumoniae*, and *E. coli* [41]. Due to the significance of LPS in the outer membrane of Gram-negative bacteria as an outer component, it maintains membrane integrity and is antibiotic penetration-resistant. The disruption of the OM has been proposed as a promising strategy by targeting LPS biosynthesis and transport, specifically the enzymes involved in both biological processes [42].

Another emerging approach targets the reduction in the robustness of MDR Gram-negative bacteria by disrupting OM lipoprotein transport and biogenesis. These proteins are acylated, anchored in the inner membrane, and transported to the OM by dedicated transport systems, such as the ATP-binding cassette (ABC) transporter LolCDE. Lipoproteins play fundamental roles in OM assembly, nutrient uptake, and resistance to environmental stress and antibiotics [43]. Consequently, pathogenic bacteria have developed decreased reliance on the lipoprotein transport pathway, which is now gaining traction as a valuable target for novel antibacterial therapies [36].

The Lol (Localization of lipoproteins) pathway constituting the LolCDE transporter is required to extract lipoproteins from the inner membrane and convey them to the OM. Key components of the Lol pathway include:The LolCDE Complex—The ATP-binding cassette (ABC) transporter LolCDE extracts lipoproteins from the inner membrane and is critical for their subsequent delivery to the OM. Recent structural studies have shed light on the transport mechanism of lipoproteins by LolCDE, involving significant rearrangements in the transmembrane helices causing large-scale “extrusion” movement of lipoprotein from the membrane [41]. Muñoz et al. [3] recently designed a novel class of selective and potential Gram-negative pathogenic bacteria active antibiotics called lolamicin [3]. For the drug design, they targeted the essential LolCDE complex, which mediates the transport of OM-specific lipoproteins to the periplasm via the cytoplasmic membrane [44]. Lolamicin displayed activity against an MDR-isolates collection of 130 strains via efficacy against multiple murine pneumonia and septicemia models, as well as protection against *C. difficile* gut infection in a mouse model of the microbiome [5]. They suggested that lolamicin is active against Gram-negative pathogenic bacteria but inactive toward commensals (both Gram-positive and -negative) based on low-sequence homology between the two classes of organisms. Thus, lolamicin seems to be at least a potential candidate for testing and possible application against human infections.Chaperones: LolA and LolB are periplasmic chaperone proteins that transfer lipoproteins across the periplasm and insert them into OM. If either LolA or LolB is disrupted, lipoproteins become mislocalized and defects in OM assembly arise [45].Inhibition of Lipoprotein Diacylglyceryl Transferase (Lgt): It catalyzes the first step in lipoprotein biogenesis, where it acylates lipoproteins in the inner membrane. Lgt inhibitors like G2824 mess up OM permeability to make bacteria more susceptible to serum killing and antibiotics. Importantly, these inhibitors show activity against strains resistant to other lipoprotein pathway inhibitors [46].

While interfering with lipoprotein transport provides a powerful approach, resistance mechanisms developed by bacteria counteract these inhibitors. Deleting the *lpp* gene, which encodes Braun’s lipoprotein in Gram-negative bacteria, circumvents inhibiting downstream steps in lipoprotein biosynthesis. On the other hand, moving down the path and targeting later steps, as with inhibitors against Lgt itself, seems less susceptible to this resistance mechanism [47], making them more favorable for clinical development.

Inhibiting lipoprotein biogenesis or transport induces bacteria’s sensitivity to current antibiotics and may repurpose previously inadequate drugs for use against MDR pathogens. This method is particularly encouraging for infections caused by *P. aeruginosa*, *A. baumannii*, and *K. pneumoniae*, which have shown resistance to traditional antibiotics [36].

Stress response systems regulate lipoprotein trafficking in bacteria. In *E. coli*, the Cpx system monitors the delivery of OM lipoproteins and activates a stress response to promote protection against the so-called trafficking defects. Targeting this system can make lipoprotein transport inhibitors more efficacious, as this is essential for bacterial survival in the presence of lipoprotein trafficking indisposition [47].

### 4.4. These Complexities Pose Their Own Challenges in Clinical Development

The potential for OM disruptors to cause toxicity is one of the key challenges in developing these findings into clinically relevant treatments. Murepavadin, an Lpt system inhibitor that showed promise in preventing *P. aeruginosa* pathogenesis, was shown to be effective but discontinued due to the observed nephrotoxicity during clinical trials [40]. This provides a rationale for why these two considerations must change—namely, all mechanisms that disrupt bacterial membranes may also adversely affect mammalian cell membranes, making it difficult to find drugs with enough specificity and safety.

Combination therapies against Gram-negative OM will be required in the future. Treatment with OM disruptors and intracellular-targeting antibiotics can synergistically increase drug efficacy while decreasing the likelihood of resistance emergence. This is also exemplified by other combination therapies that use OM perturbation to help antibiotics, such as β-lactams and glycopeptides, access their targets in Gram-negative bacteria [48]. Using efflux and permeability predictions in medicinal chemistry can enable the rational design of updated antibiotics [16]. Finally, the inhibitors of drugs that target resistance mechanisms (e.g., viz efflux pump, β-lactamase), when developing along with agents disrupting the OM, can reduce antibiotic resistance and retain current antibiotics viability, gaining time for novel alternative class antibacterial development [49].

## 5. Emerging Antibiotics and Strategies

In recent years, several new compounds and strategies have emerged. These new antibiotics include novel classes and derivatives targeting resistant pathogens, innovative delivery systems, and combination therapies (Table 1).

### 5.1. Derivatives of Tetracycline (Eravacycline)

Eravacycline, a new synthetic tetracycline derivative recently approved for treating complicated intra-abdominal infections, has broad-spectrum activity against Gram-positive and Gram-negative bacteria, including MDR strains of *A. baumannii* and *E. coli*. Eravacycline was designed to avoid the most common tetracycline resistance mechanisms, such as efflux pumps and ribosomal protection proteins [50,51].

### 5.2. Cefiderocol

Cefiderocol is a novel siderophore cephalosporin obtained from cephalosporin C (CPC) that utilizes the siderophore–iron complex pathway to penetrate the OM of Gram-negative bacteria reaching the periplasmic space and stabilize a number of β-lactamases [52]. Cefiderocol avoids some mechanisms of resistance, such as porin mutations and efflux pumps, making it highly active against carbapenem-resistant *Enterobacterales*, *P. aeruginosa, A. baumannii*, and cohorts such as cefatazidime/avibactam, ceftozolane/tazobactam, and others [53,54,55]. Although cefiderocol was considered a promising solution against most MDR Gram-negative bacteria, resistance commenced to increase in clinical practice by mechanisms not yet solved that seemingly act together [54]. More recently, Karakonstantis et al. [55] investigated the world prevalence of cefiderocol non-susceptibility (CFDC-NS) in Gram-negative pathogens and concluded that prevalence is globally low but significantly high for carbapenem-resistant *A. baumannii* in areas where this MDR pathogen is endemic.

### 5.3. Aminoglycoside Derivatives (Plazomicin)

Plazomicin is a next-generation aminoglycoside with reduced susceptibility to aminoglycoside-modifying enzymes, the most prevalent resistance mechanism to this class of antibiotics. It is indicated for complicated urinary tract infections (UTIs) and demonstrates in vitro potency against MDR *Enterobacterales*, including carbapenemase-producing strains [56,57,58].

### 5.4. Delafloxacin

Delafloxacin (DLX) is a novel anionic non-zwitterionic, broad-spectrum fluoroquinolone (FQ) acting on Gram-positive and Gram-negative bacteria such as methicillin-resistant (MRSA) and methicillin-susceptible (MSSA) *S. aureus*, some species of *Streptococcus*, *E. coli*, *P. aeruginosa*, and *K. pneumoniae* [59,60,61]. DLX inhibits DNA gyrase and topoisomerase IV, two essential type-II topoisomerases involved in bacterial replication, transcription, DNA repair, and recombination [61], and its activity is influenced by its structural properties. Thus, under acidic conditions prevailing in skin infections, DLX is protonated and shows greater in vitro activity against Gram-positive bacteria [61]. Comparison of the in vitro activity of DLX on isolates from patients suffering from acute skin infections and osteomyelitis with other antimicrobials revealed that DLX was markedly more potent than other FQs such as levofloxacin and ciprofloxacin. Mutations in *gyrA* and *parC* encoding for subunits A of DNA gyrase and topoisomerase IV, respectively, via amino acid substitution, are proposed to give rise to DLX resistance [60].

### 5.5. New Glycopeptides (Carbomycin)

Glycopeptides (GPAs) are non-ribosomal natural peptides produced by various organisms that act on the CW of Gram-positive but not Gram-negative bacteria as they are not capable of crossing the OM due to their physicochemical properties [62]. In fact, GPAs are considered by many scholars as the ultimate defense against MDR Gram-positive bacteria. Corbomycin, for instance, is a new GPA that binds the bacterial CW to inhibit autolysin-mediated peptidoglycan remodeling during bacterial growth. It has been proven effective against MRSA and other resistant Gram-positive pathogens [63]. Recently, GPAs were divided into five subclasses according to the residues present in positions 1 and 3 of the glycopeptide with corbomycin and complestatin (GP6378) belonging to subclass V [64].

### 5.6. Nanomaterials and Biomedicine

Nanotechnology has emerged as a novel approach for optimal drug delivery, targeted therapy, nano-surgery, tissue engineering, and other applications that have remarkably changed the course of medicine. In the case of antibiotics, nanoparticles (NPs) would enhance drug pharmacokinetics, improve bacterial cell internalization, and facilitate biofilm penetration. These drug delivery platforms decrease the development of resistance by improving effective targeting, leading to a lower dose requirement of antibiotics. Recent developments in nanotechnology involve using polymer-based and inorganic nanoparticles as carriers for traditional antibiotics, improving their efficacy against resistant pathogens [65]. In a recent and illustrative review, liposomes, protein nanoparticles, dendrimers (synthetic polyfunctional macromolecules), and several other nanomaterials were described as key carriers for antibiotic delivery, tissue penetration systems, and reductions in adverse effects, thereby being considered as promising in the future of nanomedicine [66].

### 5.7. Antibiotic Potentiators

Antibiotic potentiators are adjuvants that lack anti-bacterial activity but they enhance and restore activity or extend the spectrum of antibiotics with different targets when used in combination. The possible mechanisms of action of potentiators are still a matter of discussion and include the loosening of OM or inner membrane permeability, the inhibition of resistance enzymes or EPs, the alteration of vectors that carry the resistance genes, etc. [67]. For instance, MD-124 is a cationic small-molecule adjuvant with a diamidine core that re-established (in the context of current antibiotic therapies) the drug susceptibility of MDR Gram-negative bacteria to clinically relevant antibiotics. MD-124 acts synergistically with antibiotics such as colistin (polymyxin E), restoring susceptibility in carbapenem-resistant strains by selectively permeabilizing the OM [48]. The most used potentiators in clinical practice are β-lactamase inhibitors [67].

### 5.8. Bacteriophage Therapy

Viruses that infect bacteria, known as bacteriophages or bacterial killers, are an alternative to traditional antibiotics, which still need advancement via designs implementing in vivo and ex vivo-based approaches. One design that has received special attention is phage therapy, which shows promise in the fight against antibiotic-resistant bacteria, especially under personalized medicine circumstances. Phages have been used to treat drug-resistant *P. aeruginosa* infections in cystic fibrosis patients [68] and other bacterial infections. It is well documented that bacteriophages can be used not only to treat bacterial infections but also to prevent biofilm formation, which implies serious risks, particularly in the case of MDR pathogens. In this context, it has been demonstrated that phages can prevent the formation of these structures by *E. coli* and *P. aeruginosa* in biotic and abiotic matrixes, respectively [69]. Using lytic phages charged with NPs showing antibacterial properties would be expected to result in an efficient antibacterial therapy by blocking biofilm formation. Accordingly, a novel synergistic construction of T7 phages bearing silver nanoparticles (AgNPs) eliminated biofilm formation in *E. coli* and was not toxic for eukaryotic cells [69], thereby paving the way for similar combinations in infection therapies, particularly in MDR pathogens.

## 6. Post-Genomic Era Antibacterial Drug Discovery

The post-genomics era has uncovered many new targets for antibiotic drug discovery and developed strategies to combat the growing threat of resistance against current antibacterials. Three of these targets, discovered through advances in bacterial genomics and molecular biology, included bacterial signal transduction systems, lipid biosynthesis pathways, and structural components essential for bacterial viability (Table 2).

### 6.1. Histidine Kinases in Two-Component (TCS) and Quorum Sensing Systems

Bacterial two-component systems (TCSs), with histidine kinases (HKs) as integral components, play a critical role in bacterial adaptation to environmental stress, such as antibiotic exposure. Histidine kinases are involved in bacterial signaling pathways that mediate antibiotic resistance, and they can be targeted to disrupt these pathways. Due to the absence of TCSs in humans, HKs are an attractive target for the selective inhibition of bacterial growth without threat to human cells. Recent work has shown that HK inhibition can affect bacterial signaling to render pathogens less virulent. HKs possess an ATP-binding domain, which is highly conserved between bacterial species and offers a promising target for broad-spectrum antimicrobial activity [70]. Moreover, HKs also serve as the target of novel specific drugs to treat infections caused by MDR organisms, such as *S. aureus* and *A. baumannii* [71].

Other protein kinases, notably those that play roles in bacterial signaling pathways, provide attractive drug targets. Serine/threonine kinases are important pathogen virulence factors that regulate bacterial growth, and inhibitors of these enzymes in *Mycobacterium tuberculosis* may be useful candidates for treating tuberculosis. Most recent work has identified compounds that impede these kinases from their function and kill the organism in animal models [72]. A comprehensive survey of histidine kinase (HK) inhibitors has been performed, providing information on structural classifications, chemical structures, IC50 and MIC values, and antivirulence properties and exploring domains for HK targets, including the ATP-binding domain, HK sensors, the histidine phosphorylation domain, and dimerization [71].

Quorum sensing (QS) is achieved via a cell-to-cell communication system allowing Gram-positive and Gram-negative bacteria to regulate gene expression, including virulence factor production, as a function of population density. The inhibitory effect of quorum sensing disrupts these processes and leads to less bacterial pathogenicity and biofilm formation, which is also generally resistant to antibiotics. Accordingly, it has been shown that inhibiting the AgrC histidine kinase in the quorum-sensing system of *S. aureus* reduces virulence and biofilm formation [73]. The VraTSR three-component system is another important *S. aureus* system that recognizes cell wall damage after antibiotic exposure. Compounds that inhibit the VraTSR system have decreased the survival of this bacterium exposed to cell wall-targeting antibiotics and could be useful for combination therapies [74].

### 6.2. Lipid Biosynthesis Pathways

The OM of Gram-negative bacteria contains an important element, lipid A. Targeting its biosynthetic pathway allows for the OM to be compromised, making the cells more susceptible to antibiotics. Over the past eight years, significant effort has been directed toward other enzymes in lipid A biosynthesis, such as LpxC and LpxH, the second and fourth enzymes of the Raetz pathway. LpxC inhibitors, like LPC-233, are potent bactericidal agents against MDR pathogens without the cardiovascular toxicity of early LpxC inhibitors [75]. Inactivating other LpxC inhibitors has been shown to inhibit growth in clinically relevant Gram-negative pathogens, including *P. aeruginosa* and *K. pneumoniae* [76]. In addition, the inhibition of LpxH accumulates toxic lipid A intermediates and thus kills the bacteria due to an essential step in the fatty acid mobilization of lipid A biosynthesis. The inhibition of both LpxC and LpxH, which work in tandem with lipid A biosynthesis, thus provides a dual-action inhibition strategy that represents a powerful tool against MDR infections [77].

One such strategy has targeted agents interfering with lipid II, an important component of bacterial cell wall biosynthesis. An example of such a synergistic approach is the combination of Lipid II-targeting antibiotics with outer membrane-acting peptides to potentiate their activity against Gram-negative pathogens [78].

### 6.3. Protein Synthesis Inhibition (Hybrid Antimicrobial Peptides)

Since protein synthesis is a fundamental process for bacterial life, the ribosome has proven to be a successful target for antibiotic development. However, the most recent work centers on spermine-conjugated short proline-rich lipopeptides that target and block protein synthesis in *E. coli* without compromising bacterial membrane integrity. These lipopeptides exhibit broad-spectrum activity and synergy with conventional antibiotics, particularly in MDR strains [79]. Recent work has also focused on designing antimicrobial peptides that target the prokaryotic ribosome, preventing protein synthesis and showing high potency against MDR *S. aureus* and *E. coli* [78]. A novel strategy is to create antimicrobial peptides (AMPs) against pathogenic bacteria based on a tripartite structure consisting of a cell-penetrating peptide, a linker, and an amyloidogenic peptide. The first constituent allows the construction to enter the cell whereas the amyloidogenic peptide functions by congregating with bacterial proteins. For instance, two hybrids of AMPs containing the amyloidogenic sequence of ribosomal S1 protein from *P. aeruginosa* were constructed and showed activity against this pathogen [80]. Later, three synthetic peptides were prepared using the amyloidogenic sequence from *S. aureus* and showed activity against Gram-positive and Gram-negative bacteria [81]. More recently, the same group optimized and evaluated hybrid peptides in terms of their components and showed that these changes resulted in more active AMPs against a broader spectrum of pathogens [82].

### 6.4. The Bacterial Cell Wall and Membrane as Targets

The CW of pathogenic (and non-pathogenic) bacteria contains components absent in human cells, thus representing the most convenient target for antimicrobial drugs. Since the discovery of penicillin in 1928 and its availability in 1945, the CW of pathogenic bacteria has been the most exploited target of most antibiotics acting mainly on peptidoglycan synthesis and assembly. In a recent and very extensive review, Zhydzetski et al. [83] described a long list of agents against this structure in pathogenic Gram-positive bacteria and their mechanisms of action and resistance, emphasizing MDR bacteria. Here, we name some of the most relevant agents for the sake of brevity. In another context, Dias and Rauter [84] explored membrane-targeting antibacterials and grouped them into peptides and non-peptides. Peptides include AMPs, SAMPs (synthetic AMPs), gramicidin S, and caragenins, whereas non-peptides comprise reutericyclin, carbohydrate-derivatives such as aminoglycosides, macrolides, and others not classified like xanthones, quinolones, and benzophenones.

**Table 2 antibiotics-14-00404-t002:** New targets for antibacterial drug development in the post-genomic era.

Target	Description	Mechanism of Action	Key Examples/Notes
Histidine Kinases (HKs)	Part of Two-Component Systems (TCS) involved in bacterial stress response and antibiotic resistance	Inhibition of HKs disrupts bacterial signaling pathways, reducing pathogenicity and virulence.	- Targeting the ATP-binding domain of HKs has shown broad-spectrum antimicrobial activity [70] - Effective against MRSA and *A. baumannii* [71].
Quorum Sensing Systems	Bacterial communication system regulating virulence and biofilm formation.	Inhibiting quorum sensing reduces pathogenicity and biofilm formation.	- AgrC histidine kinase inhibition reduces virulence and biofilm in *S. aureus* [73]
Lipid Biosynthesis Pathways	Essential for bacterial membrane integrity in Gram-negative bacteria	Inhibition of lipid A biosynthesis weakens bacterial outer membrane, increasing susceptibility to antibiotics.	- LpxC inhibitors (e.g., LPC-233) show potent activity against MDR pathogens without toxicity [75]. - LpxH inhibition causes toxic lipid accumulation [77]
Protein Synthesis Inhibition	Critical for bacterial survival and growth.	Inhibition of ribosome function disrupts protein synthesis.	- Spermine-conjugated lipopeptides inhibit protein synthesis in *E. coli* [79] - Antimicrobial peptides target ribosomes in MDR strains [78]
Protein Kinase Inhibitors	Enzymes involved in bacterial growth and virulence, especially in *M. tuberculosis*.	Inhibition of serine/threonine kinases disrupts bacterial signaling and growth.	- Kinase inhibitors show efficacy in controlling tuberculosis [72]
Peptidoglycan and Membrane Targets	Peptidoglycan layer is essential for bacterial cell wall integrity.	Disruption of peptidoglycan weakens the bacterial cell wall, leading to bacterial death.	- Oligomeric structures disrupt the peptidoglycan layer in Gram-positive bacteria like *S. aureus* [78]

## 7. What Is New in Bacterial Genome Sequencing

The emergence of bacterial genome sequencing has transformed our perception of how bacteria function biochemically and molecularly, how they respond to antibiotics (clinical resistances), and how they reveal novel targets for new antibiotics. Whole-genome sequencing (WGS) and metagenomics have since emerged as integral components of clinical diagnostics and drug discovery. Here, we discuss some major breakthroughs in bacterial genome sequencing and their relevance for tracking antibiotic resistance and developing new antibiotics.

### 7.1. Predicting Antibiotic Resistance Using Whole-Genome Sequencing

Whole-genome sequencing is now an essential tool for predicting bacterial pathogens’ antibiotic-resistance genes (ARGs) [85]. Sequencing genomes of drug-resistant bacteria can detect ARGs, determine transmission potential, and predict resistance profiles at the clinical level:Antimicrobial Resistance Prediction: The use of WGS data to predict resistance phenotypes (particularly in Gram-negative bacteria) is currently being explored. Machine learning models developed on WGS data can predict resistance to multiple antibiotics with a high degree of accuracy, enabling real-time clinical decisions and potentially swift treatment tailored to the group’s type and required level of therapy [85]. This could provide in silico antibiograms, thereby assisting the correlative management of resistant infections.Genomic Databases for Resistance Genes: Novel tools such as BacAnt and ARTS are available for genome annotation regarding antimicrobial resistance genes (ARGs) and mobile genetic elements (MGEs), which are crucial for monitoring the transmission of resistance genes. Such bioinformatics platforms automate the detection of ARGs and MGEs, aiding epidemiological studies and resistance monitoring [86].

### 7.2. Discovery of New Antibiotics Through Genome Mining

The advent of genome sequencing technology has unraveled biosynthetic gene clusters (BGCs) responsible for generating novel secondary metabolites, some with promising antibiotic activity. Identifying previously uncharacterized BGCs in mined bacterial genomes can lead to discovering potential new antibiotic compounds:Triggering of Silent Gene Clusters: Many genes responsible for producing antibiotics remain silent during normal laboratory conditions. With improvements in genome mining approaches such as CRISPR/gene-editing technologies, these silent clusters have been induced for expression, producing, e.g., lignin or lactobacilli [17].Drug Discovery with Comparative Genomics: Over the years, based on comparative genomic approaches, highly conserved genes across bacterial species have been essential as novel drug targets. Despite the inherent redundancy in secondary metabolite gene clusters, it has been possible to connect several novel chemical structures with antibacterial activity to their respective gene clusters for rational antibiotic design [87].

### 7.3. A Novel Approach for Antibiotic Resistance Surveillance—Metagenomics

Research based on metagenomic sequencing has been performed in various environmental contexts, including hospitals, soil, and the human gut, to investigate microbial communities and resistomes (the collective set of all ARGs). Such an approach is useful for tracing the environmental sources of resistance genes, as well as how human activity drives the dissemination of antibiotic-resistant bacteria:Environmental Resistome Studies: Most ARGs are present in environmental bacteria long before they can be recovered from clinical pathogens. Methods such as metagenomics and metatranscriptomics could further increase the detection of unknown ARGs in bacteria that may not be culturable [88].Predictive Surveillance for Emerging Resistance—High-throughput genomic sequencing integrated with functional genomics enables the observation of resistance genes across space and time. Such predictive surveillance is necessary to detect and mitigate emerging resistance threats early enough [89].

### 7.4. Third-Generation Sequencing and Producing Complete Genomes

The emergence of third-generation sequencing platforms (e.g., PacBio and Oxford Nanopore), which can assemble entire bacterial genomes (including plasmids and mobile genetic elements coding for ARGs), have facilitated a better resolution to unveil the resistance mechanisms involved in MDR organisms. Such technologies address the limitations of short-read sequencing, which fails to resolve repetitive elements and insertion sequences that are key to understanding resistance mechanisms. Recent advances have created gold-standard protocols for using third-generation sequencing to assemble bacterial genomes and identify ARGs within antibiotic-resistant strains. Methods based on DNA extraction with magnetic beads followed by long-read sequencing are suitable for plasmid reconstruction that carries resistance genes [90]. They help to reconstruct the genetic context and transmission dynamics of ARGs.

## 8. Applying Genomic Advances to Improve Public Health

Genome sequencing has been applied to rapidly detect resistant pathogens and guide treatment in clinical settings. Nowadays, WGS is utilized to build resistance profiles of clinical isolates, which can help with effective antibiotic selection and circumvent broad-spectrum antibiotics. The information on resistance mechanisms available through genome sequencing of clinical isolates will allow targeted treatment regimens. It is extremely useful in the case of MDR organisms, where rapid diagnostics are critical to patient outcomes. To this end, machine-learning models leveraging these genomic data are being trained to predict antimicrobial resistance and optimize therapy for patients infected with resistant pathogens [91].

### 8.1. Monitoring Antimicrobial Resistance Using Genomics

WGS has changed AMR surveillance, making it possible to accurately discriminate the resistance gene and its passing into bacterial populations. The genotyping approach aids in the detection of new resistant variants that can best help formulate effective containing strategies:AMR Surveillance platforms in clinical settings: Several bioinformatics platforms have been developed, such as ResFinder, CARD, and MEGARes, to catalog ARGs and their dissemination through clinical and environmental samples. These enable real-time tracking of resistance mechanisms and have become an integral aspect of global surveillance [92]. In vitro activity against *P. aeruginosa* and *Citrobacter* spp. demonstrated the potential of nanopore sequencing for monitoring plasmid dissemination and resistance gene development during hospital outbreaks, which allows rapid action to stop the spread of resistant strains [93].Metagenomics applied to environmental samples and animal habitats: Metagenomic approaches for environmental and animal habitats are increasingly being used to identify reservoirs of ARGs that could transfer to human pathogens. Thus, understanding the environmental transmission routes/pathways of antibiotic resistance is essential, and surveillance programs that incorporate environmental sampling are needed to identify potential community sources [94]. Genomics is also being utilized for the detection of emerging pathogens and their antibiotic resistance profiles in populations. Projecting genomic diversity onto genotype-based diagnostics provides a more streamlined pathogen surveillance strategy than random sampling and can improve the identification of new resistance variants in pathogens like *Neisseria gonorrhoeae* [95].

### 8.2. Improving Detection and Response to Outbreaks

Outbreak management needs genomic sequencing. It enables public health authorities to help identify sources of outbreaks and quickly trace the transmission of infections:Outbreak Investigations and Responses: WGS has been important in multiple outbreak investigations, allowing strain differentiation between outbreak and sporadic isolates. This ability provides better identification of transmission chains, such as WGS-based tracking of *Staphylococcus pseudintermedius* infections in companion animals and their connections to pathogens in humans [96].Real-Time Genomic Tracking: With portable sequencing technologies like nanopore sequencing, bacterial infections can be tracked in real time from a genomic perspective. These technologies are also advantageous in resource-limited settings and public health emergencies when genomic data can be used critically to guide interventions [93].

### 8.3. The Advent of Personalized Medicine and Precision Public Health

Genomic tools have opened doors to personalized medicine in which the therapy can be adapted based on host genetics and pathogen identity. In a recent study, WGS, alongside machine learning, was employed to inform resistance-minimizing treatment optimization using patient-specific risk assessment for efficiency in the urinary tract and wound infections [97]. WGS now plays an integral role in precision public health by forecasting the antimicrobial resistance potential of clinical pathogens before therapy. Genomic predictions of resistance to key antibiotics such as meropenem and ciprofloxacin based on WGS improved treatment outcomes by ensuring appropriate patient therapy [85].

### 8.4. One Health Surveillance

The One Health (OH) approach acknowledges human, animal, and environmental health together so it can be used to battle antimicrobial resistance as part of the global public health challenges [98], focusing on consequences, responses, and actions [98]. Genomic technologies allow the interdisciplinary monitoring of pathogens and ARGs in the environment, fostering OH approaches. For example, genomic surveillance of *Salmonella* and *E. coli* animal pathogens has demonstrated considerable genetic overlap between animal and human infections, suggesting the need for integrated OH surveillance systems [99]. Integrating veterinary diagnostic data into national surveillance programs has been shown to enhance the detection of resistance trends in companion and food animals that may have significant implications for human health [98]. Recently, Hill et al. [100] proposed the “One Sample Many Analyses” (OSMA) principle within the OH approach. OSMA proposes maximizing the utility of each environmental sample by subjecting it to simultaneous biological, chemical, and genomic analyses. This multiparametric approach capitalizes on existing surveillance infrastructure to broaden both scope and efficiency [100]. A practical case study of the OSMA principle was recently implemented in which shellfish samples from the UK were analyzed concurrently for AMR genes, bacterial pathogens, and chemical contaminants to assess the use of shellfish as sentinels of environmental health [101].

## 9. AI for Antibiotic Discovery

AI has become a new tool with great potential to identify new drug targets and antibacterial agents against bacterial resistance [102,103], as this innovative approach allows us to analyze large datasets that outperform traditional methods and allow the discovery of complicated associations of biological information (e.g., genomic or transcriptomic) with chemical/physical data using learning algorithms and artificial neural networks [104]. Also, it helps to re-purpose the available drugs that were earlier developed for other diseases or infections as a treatment for drug-resistant infections [105]. For example, an MIT research team trained a deep-learning neural network to evaluate over 100 million compounds in their database [106]. Halicin, initially developed as a diabetes medication, was revealed by following this method. Halicin, a small molecule with a different backbone than traditional antibiotics, shows bactericidal activity against Gram-negative pathogens that would otherwise be resistant to all available antibiotics, like *M. tuberculosis* and *Enterobacteriaceae*. It has also proven effective against *Clostridioides difficile* and MDR *A. baumannii* in murine infection models, highlighting the advantages of deep learning approaches for novel targets and antibiotics [105].

### 9.1. Small-Molecule Antibiotics and Antimicrobial Peptides Discovery

These AI-based screening models minimize the extensive and costly in vitro experiments that prevent scientists from comprehensively surveying existing and repurposing novel compounds, facilitating the economic identification of viable antibiotics. AI has been particularly useful for optimizing the process of small molecule discovery—a mainstay in antibiotic development. Antimicrobial compounds have been identified using machine-learning models to screen large chemical libraries [107]. Novel small-molecule optimization using generative techniques such as Generative Adversarial Networks (GANs) and Variational Autoencoders (VAEs) are among some of the techniques employed to design novel small molecules specific to certain antimicrobial properties. Models that generate chemical structures based on specific activity (such as activity against a given bacterial pathogen) can optimize molecules for bioavailability and toxicity [108]. Recently, Macedo et al. [109] introduced MedGAN, a learning model employing a Wasserstein Generative Adversarial Network (GAN) combined with Graph Convolutional Networks (GCNs) to create novel quinoline-scaffold molecules. These molecules exhibited notable pharmacological properties, such as compliance with Lipinski’s Rule of Five [110], high uniqueness (95%), and significant novelty (93%). However, despite robust computational evidence, including detailed molecular graph evaluations and predictive toxicity modeling using the Tox21 model [109], the study lacked empirical validation through in vitro or in vivo assays to confirm the biological activity and therapeutic applicability of these AI-generated small molecules.

Several deep-learning models have been created to expedite AMP discoveries that are capable of predicting the antimicrobial activity of peptide sequences. The AMP predictive learning model (AMPs-Net) presented superior predictions of peptide activities compared to the traditional mechanisms and allowed the identification of new AMPs with potential strong antibacterial capabilities [111]. The pairing of AI with molecular dynamics simulations allows for discovering AMPs that show high therapeutic value and increased stability. Van Oort et al. [112] introduced AMPGAN v2, a bidirectional conditional generative adversarial network (BiCGAN) designed for rational antimicrobial peptide (AMP) generation. The AMPGAN v2 model presented a combination of generator–discriminator dynamics, where the generator synthesizes AMP candidates from learned latent representations, while the discriminator weighs their authenticity. The introduction of a conditioning vector (which encodes target microbes, mechanisms, and MIC values) allows controlled peptide generation, directly responding to predefined antimicrobial targets and mechanisms. AMPGAN v2 addresses critical barriers in AMP development, including short half-lives and toxicity concerns, optimizing peptide sequences specifically for antimicrobial properties. Zhao et al. [113] proposed two advanced computational frameworks, dsAMP and dsAMPGAN, designed to enhance the identification and generation of antimicrobial peptides (AMPs). The dsAMP model integrates convolutional neural networks (CNNs), attention mechanisms, and bidirectional long short-term memory (BiLSTM) layers, coupled with a transfer learning strategy to improve AMP classification. This combination of components substantially outperformed existing AMP classifiers, achieving greater than 95% accuracy, including sensitivity and specificity. The authors emphasize that their model’s performance accurately classified AMPs even when trained on relatively small datasets. Complementing the predictive power of dsAMP, the authors also presented dsAMPGAN, a generative adversarial network (GAN)-based approach to AMP generation. dsAMPGAN efficiently synthesizes novel AMP sequences with physicochemical properties closely resembling those of naturally occurring peptides. Computational validation revealed the similarity of GAN-generated peptides to authentic AMPs, specifically in hydrophobicity, aromaticity, net charge, and amino acid distribution. This validation underlines dsAMPGAN’s capability to mimic biological patterns required for antimicrobial activity [113].

More recently, Cao et al. [114] introduced a modeling framework named the Text-Guided Conditional Denoising Diffusion Probabilistic Model (TG-CDDPM), which utilizes a three-stage model. The first stage employs contrastive learning to establish robust correlations between textual descriptions and peptide sequences, enhancing the model’s precision in peptide design. The second stage refines these textual representations through a diffusion-based adapter, significantly improving text-to-peptide inference accuracy. Finally, the conditional diffusion probabilistic model in the third stage is pre-trained and fine-tuned to capture peptide features effectively, guided by textual inputs.

The effectiveness of TG-CDDPM was comprehensively validated against other state-of-the-art AMP generation models, including PepcVAE, HydrAMP, and AMPGAN v2, and notably outperformed these models, achieving superior scores across key predictive AMP activity metrics, including amPEP, CAMP r3, and IPPF-FE [114]

In a recent study, a deep learning model was used to mine large metagenomic datasets and built AMPSphere—an exhaustive catalog of ~900,000 non-redundant peptides, many previously untested experimentally [115]. This strategy has identified peptides with potent in vitro and in vivo activity against drug-resistant pathogens [115].

Collectively, these studies signify considerable advancements in AI-driven drug design, effectively addressing key barriers within the discovery pipeline. However, a recurring limitation across these studies is their reliance on computational validations, which, while robust, require subsequent empirical experimental validations.

### 9.2. Predicting Antibiotic Resistance

The prediction of antibiotic susceptibility phenotypes through analyses of bacterial WGSs is a promising approach to facilitate the development of targeted therapies. Models used to predict resistance in *Salmonella* spp. and *N. gonorrhoeae* have shown 90% accuracy [116]. Machine learning can discern subtle patterns in genomic datasets, identify novel genetic mutations associated with antibiotic resistance mechanisms, and help lay the groundwork for specific therapies [116].

Recently, Lee et al. [117] described a machine learning framework aimed at predicting antibiotic susceptibility in urinary tract infections (UTIs) by using routinely collected clinical and laboratory data from several hospitals for training the model collected between 2015 and 2019 and validated toward cases from 2020, ensuring a temporal separation between training and testing cohorts. The study employs logistic regression and random forest algorithms to model susceptibility to three first-line antibiotics—nitrofurantoin, amoxicillin-clavulanate, and ciprofloxacin—using an extensive set of clinical variables, prior antimicrobial exposure, and historical microbiological data. The study integrates over 100 features, from demographic and clinical data to detailed laboratory results (e.g., prior susceptibility patterns, blood test values, and patient comorbidities). In the report, the high AUC values for the random forest model, alongside strong F1 scores and predictive values, suggest that the models robustly differentiate between susceptible and resistant infections.

### 9.3. AI and Antibacterial Combination Treatments

Other models give rise to synergistic drug combinations that boost antibiotic activity. They also can analyze databases of drug interactions and resistance mechanisms and explore combinations that could be more effective in collaboration than when acting alone [118]. This is especially useful for reusing older antibiotics and other agents against resistant strains. Recent advances in system biology and artificial intelligence have made it possible to take into account metabolic pathways through which bacteria process drugs, thereby predicting and leading researchers toward the optimal combinations of drugs that will best impact bacterial growth and survival [119]. Computational models such as these facilitate the discovery of new drug pairs to lower resistance development [119]. As an example of the use of this approach, recently, Roche-Lima et al. [120] introduced a computational tool designed to predict synergistic antimalarial drug combinations named the Machine Learning Synergy Predictor (MLSyPred). The method incorporates various molecular fingerprints derived from chemical structures and uses several algorithms to construct predictive models of possible drug combination treatments on three *Plasmodium falciparum* strains. They used datasets that added up to 1540 drug combinations, originating from previous work by Mott et al. [121] and Mason et al. [122], involving direct laboratory evaluation of drug-combined effects against different *P. falciparum* strains. MLSyPred showed 45% precision, meaning nearly half of the predicted synergistic combinations were biologically confirmed by the previous data. Although modest, this validation step provides important experimental backing for computational predictions, adding credibility to the tool.

Because drug combinations have complex interactions affecting multiple bacterial pathways simultaneously, they are excellent candidates for AI-based approaches to machine learning and clinical prediction that elucidate the mechanism of action (MOA) of such drugs. By analyzing transcriptomic and proteomic data, they can predict the mechanism of action of novel and existing antibiotics. These models explain the interactions of antibiotics with bacterial targets and the cellular behaviors of bacteria [123]. A recently proposed method centered on a Multi-Feature-Based Attention Graph Convolutional Network (MFGAN) [124] combines three distinct molecular fingerprint types (MACCS, PubChem, and ECFP) with molecular graph representations. The model captures both local atomic environments and global molecular topology. This multimodal approach enhances the ability to identify complex structure–activity relationships. The predictive model was trained and validated on two publicly available datasets of molecular structure representations and corresponding growth inhibition data for *E. coli* and *A. baumannii* [124]. These datasets were compiled from experimental results and provide a robust basis for computational modeling. However, the nature of these datasets is retrospective, serving as in silico proxies for antibacterials. To substantiate the model’s utility in drug discovery, experimental validation is essential.

### 9.4. Advantages and Challenges

AI provides enormous value for drug development mainly due to its speed and cost-efficiencies. It analyzes vast amounts of data quickly and can find drug candidates significantly faster than conventional methods, thereby decreasing time and cost in the drug development cycle. However, the amount and quality of training data significantly restricts its effectiveness, showing that these models may be less accurate when utilizing new biological systems [118]. Although antibiotic candidates are highly predictive in the models, it is crucial to experimentally validate candidate molecules and demonstrate that they retain their efficacy and safety. To fulfill the promise of computational predictions, we require microbiologists and computational scientists to work together closely so that laboratory testing can provide insights into and confirm predictions. Lastly, many models (especially learning algorithms) are often seen as “black boxes, “ making explaining how they generate predictions challenging. Higher transparency and interpretability of the models are essential for wide acceptance in antibiotic discovery [111].

## 10. Decreased Industry Investment in Antibacterial Drug Development

Even with all the academic research efforts working toward improvements in antibacterial therapies, the lack of interest from the pharmaceutical industry in investing in the development of new antibacterial drugs is a rather significant hurdle in the fight against the increasing crisis of antimicrobial resistance. This has been related to a variety of factors, including financial, scientific, and regulatory barriers, plus the poor economic model of antibiotic development.

### 10.1. Unprofitable Antibiotics

Compared to drugs for chronic diseases, antibiotics have generally been viewed as less profitable. Antibiotics are typically provided over a short term, making them less suited for long-term revenue. In contrast, drugs for chronic diseases provide steady streams of income as patients will take the drugs for life. In addition, new antibiotics are being used as last-line treatments, effectively lowering their market potential [20]. These are economic realities that discourage pharmaceutical companies from investing in antibiotics R&D, especially when taking into account the risks associated with the discovery and approval of these products [125].

### 10.2. Challenges in Science and Technology

The drug discovery of low-hanging fruits has mostly been exploited, so it is becoming increasingly difficult to identify novel compounds with broad efficacy against multiple targets. Moreover, bacteria develop resistance mechanisms on a regular basis, shrinking the therapeutic window available for recently discovered antibiotics before resistance arises [126]. The complexity of bacterial resistance also makes developing effective antibiotics even more difficult. Bacteria such as *P. aeruginosa* and *A. baumannii* harbor various layers of resistance like efflux pumps, biofilm formation, and enzymatic degradation of drugs. These contribute to the challenge of identifying new compounds or drug combinations that can circumvent or overcome mechanisms of resistance [127].

### 10.3. Obstacles in Regulation and Market

The regulatory environment for antibiotics is affecting developers more heavily. Finally, the costs and time to produce clinical trial data are significant barriers, especially concerning antibiotics. Regulatory agencies often demand that extensive clinical trials be performed to demonstrate the efficacy and safety of a drug against resistant strains, potentially involving special patient populations that are challenging to recruit. Moreover, traditional non-inferiority trial designs for antibiotic development do not always deliver conclusive evidence of efficacy against resistant pathogens, and this complicates the approval process [128]. Additionally, profitability after market introduction is also uncertain, as antibiotic stewardship programs promote the judicious use of new antibiotics in order to defeat resistance, limiting potential sales. Preserving the effectiveness and urgency of antibiotics versus potential profitability for drug companies creates a pressure imbalance deterring innovation [129].

Creating financial incentives to make antibiotic R&D more attractive to pharmaceutical companies has been one of the main areas of efforts to revive the antibiotic pipeline. Various mechanisms have been proposed and implemented in some territories, including publicly funding early-stage research (push incentives) and market entry rewards (pull incentives). For example, US legislation such as the Generating Antibiotic Incentives Now (GAIN) Act has offered financial rewards and longer exclusivity periods for companies working on new antibiotics [19], yet these incentives were little more than band-aids applied to patients with declining investment vital signs.

New antibiotics for resistant pathogens tend to be derivatives of existing compounds rather than new chemical entities, thus primarily offering incremental health benefits [130]. In addition, the lack of a long-term sustainable business model for antibiotics has deterred pharmaceutical companies from participating in these programs. Potential solutions include innovative financial models that reward companies based on the efficacy of their antibiotics to maintain low levels of resistance (e.g., “antibiotic susceptibility bonus”) [129].

Due to the crisis of global antibiotic resistance, effective international collaboration is needed to pool resources for antibiotic funding, regulation, and stewardship initiatives [131]. Initiatives like the Wellcome Trust’s Drug-Resistant Infections Priority Program aim to develop a sustainable ecosystem for antibiotic R&D and promote public–private partnerships to provide financial incentives for sustainable, long-term investment in the R&D of new antibiotics [125] (Figure 1).

## 11. Conclusions

The emergence of bacterial infections and MDR organisms is driven by a combination of factors, including antibiotic abuse and misuse, HGT, environmental contamination, and hospital dynamics. To respond to this worldwide threat, action is needed to ensure the appropriate use of antibiotics, enhance infection control practices, and reduce the transmission of bacteria-carrying resistance genes in healthcare, agriculture, and the environment. New antibiotics acting against bacterial growth and survival mechanisms may provide further benefits over classical antibiotics, such as improved activity against resistant pathogens, reduced toxicity, and the potential for resistance development to be delayed. Focusing either on previously unused bacterial processes or combination therapies, these new approaches underline the hope of circumventing the global antibiotic resistance crisis. However, difficulties remain with targeting the Gram-negative OM in bacterial pathogens, including permeability barriers to macromolecules and efflux pumps that extrude small molecules to prevent their intracellular activity or even cytotoxicity toward bacteria. Determining better OM assembly systems, optimizing drug design, and combining therapies are pushing the boundaries toward overcoming these barriers.

In this post-genomic era, with the structure–function-based characterization of a plethora of novel molecular targets such as histidine kinases, protein and lipid biosynthesis enzymes, quorum-sensing systems, and DNA gyrase, such targets provide avenues to generate new therapeutics that evade recognized drug resistance mechanisms and can be used against MDR organisms with broader utility.

Whole-genome and metagenomic methods are central to predicting resistance, tracking its dissemination, and discovering new antibiotic agents. The ongoing development of bioinformatics tools and third-generation sequencing platforms support such tasks. In addition, these strategies can give public health agencies the ability to respond promptly to new threats, forecast resistance, and customize interventions. They will be crucial to tackling the increasing threat of antibiotic resistance and ultimately improving health at a global level with public health frameworks that integrate these technologies.

AI is transforming antibiotic discovery by providing capabilities for novel compound identification, resistance mechanism prediction, and drug combination optimization. However, it continues to face challenges, especially around data availability and model explainability. Also, the capability of these tools and databases is still incipient and essentially predictive. Undoubtedly, these protocols will be consolidated in the near future, thereby contributing to the urgent need for new and more potent antibacterials.

Economics, regulatory pathways, and scientific hurdles are the main reasons why the pharmaceutical industry has receded from developing antibiotics. Antibiotic treatments consist of a few doses over short periods of time, which means they have a very slim profit margin, even when such drugs are costly to develop, which makes companies reluctant to engage in antibiotic R&D in the first place. Although some financial incentives, such as market entry rewards and longer exclusivity periods, were introduced in 2005 to try to spur innovation using the profit motive, these developments have signified little actual successful innovation.

There is no panacea for developing new antibacterial drugs; a multifaceted approach, including resistance-resistant strategies, reviving older drugs, targeting bacterial metabolism, utilizing advanced technologies such as nanotechnology and systems biology, etc., will be essential to stave off this public health crisis for the foreseeable future. New strategies bring hope in the fight against these multidrug-resistant pathogens but will need more research, funding, and interdisciplinary cooperation to succeed (Figure 1).

## Figures and Tables

**Figure 1 antibiotics-14-00404-f001:**
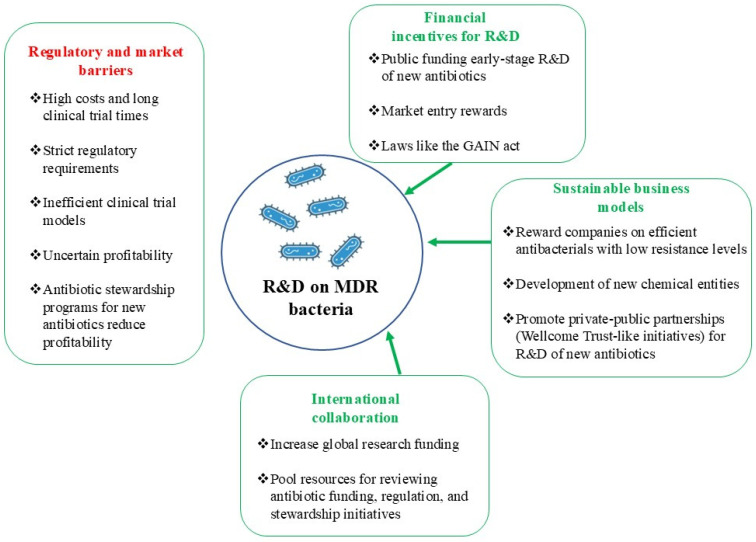
Recommended strategies to combat multi-drug resistant (MDR) bacteria.

**Table 1 antibiotics-14-00404-t001:** New antibiotics and strategies.

Antibiotic/Strategy	Class/Type	Target Pathogens	Mechanism of Action	Key Notes
Eravacycline	Tetracycline derivative	Gram-positive and Gram-negative bacteria, including *A. baumannii*, *E. coli*	Overcomes tetracycline resistance mechanisms like efflux pumps and ribosomal protection proteins	Broad-spectrum activity against MDR bacteria. Approved for complicated intra-abdominal infections.
Cefiderocol	Siderophore cephalosporin	Carbapenem-resistant *Enterobacterales*, *P. aeruginosa*	Uses bacterial iron transport system (siderophore mechanism) to penetrate Gram-negative bacteria and bypass resistance mechanisms.	Effective against Gram-negative bacteria.
Plazomicin	Aminoglycoside derivative	MDR *Enterobacterales*, including carbapenemase producers	Evades aminoglycoside-modifying enzymes	Approved for complicated UTIs.
Delafloxacin	Fluoroquinolone	Gram-positive and Gram-negative bacteria, including MRSA	Increased potency in acidic environments; effective for skin and soft tissue infections.	Effective in acidic environments found at infection sites.
Corbomycin	Glycopeptide	Gram-positive pathogens, including MRSA	Binds to bacterial cell wall, preventing autolysin activity, which differs from traditional glycopeptides like vancomycin.	Unique mechanism with low resistance development.
Nanomaterials	Nanotechnology-based strategy	Various resistant pathogens	Nanoparticles improve antibiotic delivery, increase bacterial internalization, and enhance biofilm penetration.	Enhances antibiotic effectiveness and reduces dosage requirements.
MD-124	Antibiotic potentiator	Drug-resistant Gram-negative bacteria	Permeabilizes bacterial outer membrane, enhancing antibiotic efficacy (e.g., colistin).	Potentiators offer a cost-effective strategy to combat resistance.
Bacteriophage Therapy	Phage therapy	Antibiotic-resistant bacteria (*P. aeruginosa* in CF patients)	Use bacteriophages (viruses) to target and destroy specific bacterial pathogens.	Alternative to traditional antibiotics, with promising results in personalized medicine approaches.

## Data Availability

No new data were created or analyzed in this study.

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
