# Peer review of "Novel Antibacterial Approaches and Therapeutic Strategies"

_antibiotics, 2025, doi:10.3390/antibiotics14040404_

Round 1

Reviewer 1 Report

Comments and Suggestions for Authors

The article summarizes the factors leading to antibiotic resistance and provides an insight into treatment approaches and new targets for antibiotic discovery. The article should be accepted for publication with minor changes as specified below.

 Recommendation/comments

  1. Resistance is reported against cefiderocol and delafloxacin although these antibiotics have been recently introduced. Some references are included below which should be reflected in the article and I would suggest more extensive search to keep the article updated.
  2. In vitro resistance development gives insights into molecular resistance mechanisms against cefiderocol
  3. The emergence of cefiderocol resistance in Pseudomonas aeruginosafrom a heteroresistant isolate during prolonged therapy
  4. High-level delafloxacin resistance through the combination of two different mechanisms in Staphylococcus aureus
  5. The chemical structures of the molecules discussed in the article should be included with numbers in the order in which they appear and would be more impactful to the readers.

Author Response

The article summarizes the factors leading to antibiotic resistance and provides an insight into treatment approaches and new targets for antibiotic discovery. The article should be accepted for publication with minor changes as specified below.

 Recommendation/comments

  1. Resistance is reported against cefiderocol and delafloxacin although these antibiotics have been recently introduced. Some references are included below which should be reflected in the article and I would suggest more extensive search to keep the article updated.
  2. In vitro resistance development gives insights into molecular resistance mechanisms against cefiderocol
  3. The emergence of cefiderocol resistance in Pseudomonas aeruginosafrom a heteroresistant isolate during prolonged therapy
  4. High-level delafloxacin resistance through the combination of two different mechanisms in Staphylococcus aureus
  5. The chemical structures of the molecules discussed in the article should be included with numbers in the order in which they appear and would be more impactful to the readers.

We agree with the comments 1 to 4 of reviewer 1. The article has been extended accordingly (see lines 362 to 375, 384-397, changes highlighted in green).

Regarding point 5, we felt that including the molecule structures as suggested would require extending this section to discuss the mechanisms of action of each molecule into a more focused review on that particular matter, which is beyond the scope of the present review since our main objective is to give an overview of the current situation of antibiotics resistance in regard to the general mechanism of action, mechanism of resistance and novel strategies being developed to counter the antibiotics resistance, a perspective that we believe is particularly valuable for a diverse readership that includes not only experts but also students (both graduate and undergraduate) and early-career researchers seeking to enter the field. Other sections have been extended further at the suggestion of the reviewers (changes highlighted in green throughout the whole article), which we think have improved the article’s main aim (included as a final paragraph at the end of the introduction (see lines 77 to 82).

Reviewer 2 Report

Comments and Suggestions for Authors

Novel Antibacterial Approaches and Therapeutic Strategies

The review goes antibiotics resistance issues in regard to mechanism of action, mechanism of resistance and novel strategies being developed to counter the antibiotics resistance. Along with the experimental advancements, review tries to cover AI-based advancements in the field of antibiotics treatments.

Recommendation: The authors have a done a very good job of covering various experimental advancements in the field and discuss the various new types of drug derivatives. But, authors fail to provide examples of newly developed computational tools.

I believe authors should discuss more AI tools with applications in the small molecule generation, antimicrobial peptide predictions.

Here are some of the examples of the articles that authors can discuss to improve the quality of this review.

  1. Small molecule generation and peptide predictions: PMID: 38265916, 38216614, 38791544, 39452213 and TG-CDDPM, AMPGAN. There are many more such recently published tools with high accuracies that authors can search and discuss.

  1. The section 9.4 and 9.5 are very vague and they should be modified to reflect more details. Section 9.4 can use an example PMID: 38165555 which talks about the antimalarial drug synergy prediction.

Note: Other than these suggestions, I found the review very engaging and full of information.

Comments on the Quality of English Language

The language in section 9.4 is very vague and can be changed to improve readability.

Author Response

The review goes antibiotics resistance issues in regard to mechanism of action, mechanism of resistance and novel strategies being developed to counter the antibiotics resistance. Along with the experimental advancements, review tries to cover AI-based advancements in the field of antibiotics treatments.

Recommendation: The authors have a done a very good job of covering various experimental advancements in the field and discuss the various new types of drug derivatives. But, authors fail to provide examples of newly developed computational tools.

I believe authors should discuss more AI tools with applications in the small molecule generation, antimicrobial peptide predictions.

Here are some of the examples of the articles that authors can discuss to improve the quality of this review.

  1. Small molecule generation and peptide predictions: PMID: 38265916, 38216614, 38791544, 39452213 and TG-CDDPM, AMPGAN. There are many more such recently published tools with high accuracies that authors can search and discuss.

  1. The section 9.4 and 9.5 are very vague and they should be modified to reflect more details. Section 9.4 can use an example PMID: 38165555 which talks about the antimalarial drug synergy prediction.

Note: Other than these suggestions, I found the review very engaging and full of information.

Comments on the Quality of English Language

The language in section 9.4 is very vague and can be changed to improve readability.

We thank the reviewer for their positive feedback on the review. We have expanded the subsection on AI tools to accommodate the reviewer's suggestions, including most of the references pointed (see lines 728– 737, highlighted in green). We also decided to combine sections 9.1 (Small molecules antibiotic) and 9.2 (Discovery of antimicrobial peptides (AMP)) into one single section (9.1- Small-molecule Antibiotics and Antimicrobial Peptides Discovery, line 717) since it makes a more fluent text. Sections 9.4 and 9.5 were also combined (now 9.3 AI and Antibacterial Combination Treatments, line 813), and the text was expanded to accommodate the suggestion of the reviewer and more information to reflect more details and make it less vague as suggested (see lines 820 to 849).

Reviewer 3 Report

Comments and Suggestions for Authors

In this review, Nino-Verga et al. attempt to provide a comprehensive review of all novel antibacterial strategies today. Unfortunately, I cannot recommend publication only because I found that the review is too broad and does not provide the depth that I look for when I am reading a review of the literature. Instead of this review, I would prefer going to another more focused review that can give me a deeper grasp of the literature and its applications.

For example, in Section 4, the authors provide a three page summary of the literature involving challenges in developing antibiotics targeting gram-negative outer membranes. However, there is a much more detailed review that covers this topic in greater depth: Kumari and Saraogi, “Navigating Antibiotic Resistance in Gram-Negative Bacteria: Current Challenges and Emerging Therapeutic Strategies,” Chemphyschem, published ahead of print on February 19, 2025.

Similarly there is a solid and very recent review by Bilal et al. on the role of AI and machine learning in predicting and combating antimicrobial resistance: “The role of artificial intelligence and machine learning in predicting and combating antimicrobial resistance,” Comput Struct Biotechnol J 18 (2025):423-439. It is much more comprehensive than the discussion found in Section 9 of this manuscript. I would go to that paper over this manuscript.

In the end, I cannot justify recommending publication of this review because I do not know how it adds value to the existing literature. May I suggest that the authors focus their review -- say on Sections 7, 8, and 10 -- to get a tighter and more focused work that highlights the role of genome sequencing and NGS on antibacterial therapeutic strategies.

Author Response

In this review, Nino-Verga et al. attempt to provide a comprehensive review of all novel antibacterial strategies today. Unfortunately, I cannot recommend publication only because I found that the review is too broad and does not provide the depth that I look for when I am reading a review of the literature. Instead of this review, I would prefer going to another more focused review that can give me a deeper grasp of the literature and its applications.

For example, in Section 4, the authors provide a three page summary of the literature involving challenges in developing antibiotics targeting gram-negative outer membranes. However, there is a much more detailed review that covers this topic in greater depth: Kumari and Saraogi, “Navigating Antibiotic Resistance in Gram-Negative Bacteria: Current Challenges and Emerging Therapeutic Strategies,” Chemphyschem, published ahead of print on February 19, 2025.

Similarly there is a solid and very recent review by Bilal et al. on the role of AI and machine learning in predicting and combating antimicrobial resistance: “The role of artificial intelligence and machine learning in predicting and combating antimicrobial resistance,” Comput Struct Biotechnol J 18 (2025):423-439. It is much more comprehensive than the discussion found in Section 9 of this manuscript. I would go to that paper over this manuscript.

In the end, I cannot justify recommending publication of this review because I do not know how it adds value to the existing literature. May I suggest that the authors focus their review -- say on Sections 7, 8, and 10 -- to get a tighter and more focused work that highlights the role of genome sequencing and NGS on antibacterial therapeutic strategies.

We thank Reviewer 3 for their thoughtful feedback. While we acknowledge the value of more focused reviews, our article's broad scope is purposeful and distinctive. Its integrative approach brings together a range of emerging and antibacterial strategies, presenting a perspective that we believe is particularly valuable for a diverse readership that includes not only experts but also students (both graduate and undergraduate) and early-career researchers seeking to enter the field. We strongly believe that the review, with its broad structure, would serve as an accessible nonetheless rigorous overview of current innovations in antibacterial therapy. By connecting advances across multiple domains, it provides a consolidated resource that supports cross-disciplinary learning and fosters a broader understanding of the difficulties and opportunities in addressing antimicrobial resistance (AMR). We have aimed to highlight not only the advances within individual strategies but also the way these approaches intersect, complement one another, and collectively contribute to the fight against AMR. We expect that this explanation clarifies our rationale for preserving the article’s current structure. We are confident that the revised manuscript - with a clarifying statement as the last paragraph of the introduction to make our intent more explicit (highlighted in green in the article, see lines 76 to 82) - aligns with the aims of the journal, and will be of value to its wide range readership. We are grateful to the reviewer for the opportunity to further articulate the purpose and contribution of our work, and we remain open to any additional suggestions that could help strengthen the manuscript.

Reviewer 4 Report

Comments and Suggestions for Authors

The manuscript presents a comprehensive discussion on the growing threat of antimicrobial resistance (AMR) and explores novel strategies to address this global health challenge. The review does a commendable job of integrating multiple perspectives, such as genomic approaches, artificial intelligence, and economic factors affecting antibiotic development. At the same time it would benefit from a clearer identification of novel insights or underexplored avenues for future research to enhance its originality. The manuscript is generally well-structured, with a logical progression from the underlying causes of AMR to possible solutions. If the authors can refine the presentation and expand on the discussion of innovative strategies, this article has the potential to make a valuable contribution to the field. I believe that the manuscript can be published after substantial revision and content enhancement.

Major Comments:
1) The review would significantly benefit from the addition of a schematic figure illustrating the authors' proposed strategies for continued research and development (R&D), industry-academic partnerships, and financial mechanisms to combat multidrug-resistant (MDR) bacteria.

2) Some sections contain subsections that, in turn, discuss only a single study (with only one cited reference). This does not adequately reflect diverse perspectives or sufficiently represent recent advancements in the field. Please consider incorporating additional relevant studies and discussing them in the manuscript where necessary. Examples of such subsections are listed below.

Lines 188-195. “4.1. The outer membrane as a permeability barrier” 1)

Lines 197-203. “4.2. Porins and Antibiotic Entry”

Lines 204-209. “4.3. Efflux Pumps “.

Lines 299-308. “4.6. Chemical Space Limitations”. ─ This section also contains only a single reference (a 2020 study). As an alternative, I suggest reviewing other relevant works and expanding this section accordingly. For example, https://doi.org/10.1016/j.bmcl.2024.129844 , https://doi.org/10.1038/s41598-022-12376-1 , etc.

Lines 361-366. “5.5. New β-Lactam/β-Lactamase Inhibitor Combinations (Meropenem-Vaborbactam)” ─ Unfortunately, I could not find a discussion of the Meropenem-Vaborbactam combination in the cited reference [13]. Please consider more detailed studies on this topic, such as: https://doi.org/10.3390/ijms25179574 or https://doi.org/10.2147/IDR.S150447 , etc.

Lines 368-373. “5.6. New Glycopeptides (Corbomycin)”
Lines 375-382. “5.7. Nanomaterials in the Context of the Drug Delivery”

Lines 384-389. “5.8. MD-124: Antibiotic Potentiator”

Lines 391-398. “5.9. Bacteriophage Therapy”

Lines 459-467. “6.3. Protein Synthesis Inhibition” ─ Additionally, please consider studies that evaluate antimicrobial peptides derived from bacterial ribosomal proteins: https://doi.org/10.3390/ijms23010524 , https://doi.org/10.3390/ijms22189776 , etc.

Lines 469-476. “6.4. Inhibitors of DNA Gyrase and Topoisomerase”.

Lines 478-482. “6.5 Targets in Bacterial Peptidoglycan and Membrane.”.

Lines 614-624. “8.4. One Health Surveillance”.

Lines 686-692. “9.5. Mechanism of Action (MOA) Prediction”.

Author Response

The manuscript presents a comprehensive discussion on the growing threat of antimicrobial resistance (AMR) and explores novel strategies to address this global health challenge. The review does a commendable job of integrating multiple perspectives, such as genomic approaches, artificial intelligence, and economic factors affecting antibiotic development. At the same time it would benefit from a clearer identification of novel insights or underexplored avenues for future research to enhance its originality. The manuscript is generally well-structured, with a logical progression from the underlying causes of AMR to possible solutions. If the authors can refine the presentation and expand on the discussion of innovative strategies, this article has the potential to make a valuable contribution to the field. I believe that the manuscript can be published after substantial revision and content enhancement.

Major Comments:
1) The review would significantly benefit from the addition of a schematic figure illustrating the authors' proposed strategies for continued research and development (R&D), industry-academic partnerships, and financial mechanisms to combat multidrug-resistant (MDR) bacteria.

2) Some sections contain subsections that, in turn, discuss only a single study (with only one cited reference). This does not adequately reflect diverse perspectives or sufficiently represent recent advancements in the field. Please consider incorporating additional relevant studies and discussing them in the manuscript where necessary. Examples of such subsections are listed below.

Lines 188-195. “4.1. The outer membrane as a permeability barrier” 1)

Lines 197-203. “4.2. Porins and Antibiotic Entry”

Lines 204-209. “4.3. Efflux Pumps “.

Lines 299-308. “4.6. Chemical Space Limitations”. ─ This section also contains only a single reference (a 2020 study). As an alternative, I suggest reviewing other relevant works and expanding this section accordingly. For example, https://doi.org/10.1016/j.bmcl.2024.129844 , https://doi.org/10.1038/s41598-022-12376-1 , etc.

Lines 361-366. “5.5. New β-Lactam/β-Lactamase Inhibitor Combinations (Meropenem-Vaborbactam)” ─ Unfortunately, I could not find a discussion of the Meropenem-Vaborbactam combination in the cited reference [13]. Please consider more detailed studies on this topic, such as: https://doi.org/10.3390/ijms25179574 or https://doi.org/10.2147/IDR.S150447 , etc.

Lines 368-373. “5.6. New Glycopeptides (Corbomycin)”
Lines 375-382. “5.7. Nanomaterials in the Context of the Drug Delivery”

Lines 384-389. “5.8. MD-124: Antibiotic Potentiator”

Lines 391-398. “5.9. Bacteriophage Therapy”

Lines 459-467. “6.3. Protein Synthesis Inhibition” ─ Additionally, please consider studies that evaluate antimicrobial peptides derived from bacterial ribosomal proteins: https://doi.org/10.3390/ijms23010524 , https://doi.org/10.3390/ijms22189776 , etc.

Lines 469-476. “6.4. Inhibitors of DNA Gyrase and Topoisomerase”.

Lines 478-482. “6.5 Targets in Bacterial Peptidoglycan and Membrane.”.

Lines 614-624. “8.4. One Health Surveillance”.

Lines 686-692. “9.5. Mechanism of Action (MOA) Prediction”.

  • We thank the reviewer for his comments and suggestions. We agree to include the suggested figure (now Fig. 1, see line 929).
  • Thank you for your comments. We have edited the article, expanding and deepening the highlighted sections and added additional references. Some subsections have been merged to provide a more integrated and fluid reading, so we cannot provide a point-by-point response to every subheading pointed out by the reviewer (see lines 186 to 240, 362 to 375, 515 to 548, 691 to 699, 820 to 849, highlighted in green). We hope that the edited article meets the reviewer's requirements.

It is important to point out that, as mentioned to the other reviewers, our main objective is to give an overview of the current situation of antibiotic resistance in regard to the general mechanism of action, mechanism of resistance, and novel strategies being developed to counter the antibiotics resistance, a perspective that we believe is particularly valuable for a diverse readership that includes not only experts but also students (both graduate and undergraduate) and early-career researchers seeking to enter the field. Other sections have been extended further at the suggestion of the reviewers (changes highlighted in green throughout the whole article), which we think have improved the article’s main aim (included as a final paragraph at the end of the introduction (see lines 77 to 82).

Round 2

Reviewer 3 Report

Comments and Suggestions for Authors

I still have concerns about such a broad review but I will defer to the decision of the editors and accept the manuscript.

Reviewer 4 Report

Comments and Suggestions for Authors

I have no comments.